# Brain Glucose Metabolism and *COMT* Val 158 Met Polymorphism in Female Patients with Work-Related Stress

**DOI:** 10.3390/diagnostics14161730

**Published:** 2024-08-09

**Authors:** Saga Steinmann Madsen, Thomas Lund Andersen, Jesper Pihl-Thingvad, Lars Brandt, Birgitte Brinkmann Olsen, Oke Gerke, Poul Videbech

**Affiliations:** 1Center for Neuropsychiatric Depression Research, Mental Health Center Glostrup, University of Copenhagen, 2600 Glostrup, Denmark; saga.christa.steinmann.madsen@regionh.dk (S.S.M.); videbech@dadlnet.dk (P.V.); 2Research Unit of Clinical Physiology and Nuclear Medicine, Department of Clinical Research, University of Southern Denmark, 5230 Odense, Denmark; 3Department of Nuclear Medicine, Odense University Hospital, 5000 Odense, Denmark; 4OPEN (Open Patient data Explorative Network), Odense University Hospital, 5000 Odense, Denmark; 5Department of Clinical Physiology, Nuclear Medicine and PET, University Hospital Rigshospitalet, 2100 Copenhagen, Denmark; 6Department of Occupational and Environmental Medicines, Odense University Hospital, 5000 Odense, Denmark; jesper.pihl-thingvad@rsyd.dk (J.P.-T.);; 7Research Unit of Occupational & Environmental Medicine, Department of Clinical Research, University of Southern Denmark, 5000 Odense, Denmark; 8Department of Surgical Pathology, Zealand University Hospital, 4000 Roskilde, Denmark

**Keywords:** catechol-O-methyltransferase, cerebral, FDG, occupational stress, psychosocial work environment

## Abstract

Stress is a ubiquitous challenge in modern societies. Symptoms range from mood swings and cognitive impairment to autonomic symptoms. This study explores the link between work-related stress and the neurobiological element of brain processing, testing the hypothesis that patients with occupational stress have altered cerebral glucose consumption compared to healthy controls. The participants’ present conditions were evaluated using an adapted WHO SCAN interview. Neural activity at rest was assessed by positron emission tomography (PET) with the glucose analogue [18F]fluorodeoxyglucose. Participants were genotyped for the Val158Met polymorphism of the *COMT* gene, believed to influence stress resilience. This study included 11 women with work-related stress and 11 demographically comparable healthy controls aged 28–62 years, with an average of 46.2 years. The PET scans indicated clusters of decreased glucose consumption primarily located in the white matter of frontal lobe sub-gyral areas in stress patients. *COMT* Val158Met polymorphism detection indicated no immediate relation of the homozygous alleles and stress resilience; however, healthy controls mainly had the heterozygous allele. In conclusion, the results support that work-related stress does affect the brain in the form of altered glucose metabolism, suggesting neurobiological effects could be related to white matter abnormalities rather than gray matter deterioration. Genotyping indicates a more complex picture than just that of the one type being more resilient to stress. Further studies recruiting a larger number of participants are needed to confirm our preliminary findings.

## 1. Introduction

The overall purpose of this study was to investigate possible differences between stress patients and healthy controls in the perceived state of their mental health situation, somatic symptoms of stress, voxel-wise glucose metabolism in the brain, and *COMT*-genotype variation. Hence, this study aimed to clarify the symptoms reported by stress patients and to implement an exploratory setup to investigate deviations in baseline brain function measured by cellular glucose consumption, with a focus on the field of nuclear medicine using positron emission tomography (PET) supported by magnetic resonance imaging (MRI). Additionally, all participants were tested for genotype variation in the *COMT* gene, known as the Warrior/Worrier gene.

### 1.1. Background

Mental disorders pose one of the greatest challenges to modern societies today, and stress with comorbidities is among the costliest disorders in Denmark, due to both health issues and being a major cause of lost earnings in the corporate sector, and the prevalence is increasing [1,2]. Danish and international studies [3,4] clearly show a significant correlation between psychosocial stressors in the work environment and the development of mental disorders, increasing the risk of depression, dementia, cardiovascular disease, and risk of accidents [5]. Multiple studies detail the etiology behind work-induced mental disorders and the prevailing stressors that cause these conditions [6,7], with the most pertinent factors being high job demands, high emotional demands, organizational injustice, low support, and decision authority [8]. Stress is correlated with the executive and emotional functions of the human brain as well as with physiological functions. However, to the best of our knowledge, there is very little research linking psychosocial work stressors to alterations in the energy metabolism of the brain. 

Despite the high prevalence of stress, there is no clear definition of the disorder. The term “stress” is used relatively broadly in everyday life, applied to everything from busyness to mental and physical symptoms requiring hospitalization. Work-related stress is also referred to as occupational stress. Prolonged stress reactions are often diagnosed as adjustment disorders, F43.2 in the ICD-10, due to a lack of a more specific diagnose [9]. However, the choice of diagnosis varies from country to country and within national borders, as described by Kristiansen et al. and Rotvig et al. [10,11]. F43.8 “Other reactions to severe stress” and F43.9 “Reaction to severe stress, unspecified” are other diagnostic codes applied to stress conditions. Stress covers a wide spectrum of symptoms with an overlap with depression and the anxiety spectrum. In general, the condition is referred to as occupational stress or just stress, and of these terms, the latter will be applied in this article. Adjustment disorder is defined as a maladaptive reaction to an identifiable psychosocial stressor or multiple stressors, in this study related to work. The disorder is characterized by preoccupation with the stressor/stressors or its consequences, including excessive worry, recurrent and distressing thoughts about the stressor, or constant rumination about its implications, as well as failure to adapt to the stressor, which causes significant impairment in personal, family, social, educational, occupational, or other important areas of functioning. Moreover, it should not be possible for the symptoms be to explained better by another mental disorder, e.g., PTSD, Mood Disorder, or another disorder specifically associated with stress. By definition, the symptoms should typically resolve within six months, unless the stressor persists for a longer duration [9]. From its earliest conception, the stress response has been linked to activations of the hypothalamic–pituitary–adrenal axis and the sympathetic nervous system [12]. However, little is currently known about how the mechanisms of stress influence the brain and especially the higher brain functions based in the prefrontal cortex, such as executive functions and emotional regulation measured by glucose consumption.

### 1.2. Brain Imaging

Today, technological advancements offer a variety of means and modalities for studying the brain, which can be applied to address different aspects of brain anatomy and function. In this study, we aimed to use a PET/MRI scanner to produce whole-brain images. From a neuroscientific perspective, both PET and MRI imaging play important roles in understanding the brain’s structure, physiology, and function.

The main modality in this study is based in the field of nuclear medicine, specifically the PET modality. Brain functions are directly linked to glucose metabolism [2,13]. As part of the PhD project *Neurobiological Effects of Work-Related Stress* [14], this observational comparative study investigated global brain glucose consumption in the resting state with the 2-[18F]fluoro-2-deoxy-D-glucose (FDG) radiotracer using PET. Images were acquired on a combined GE Signa 3T PET/MR scanner (GE Healthcare, Waukesha, WI, USA).

#### 1.2.1. Principles of PET

PET imaging is based on the principle of radioactivity tracer imaging, which involves injecting a radioactive tracer into the bloodstream [15]. Radiotracers are substances labeled with radioactive unstable isotopes. These isotopes undergo radioactive decay over time, emitting positrons (e^+^), which interact with electrons (e^−^) in biological systems, which ultimately leads to two annihilation photons detected in a PET scanner. Detectors in the scanner capture annihilation photons to create images that, depending on the tracer, detect biochemical features of physiological parameters (Figure 1). 

The advantages of PET imaging lie in the radiotracers, which can each be designed to target specific biological processes or molecules, such as the measurement of metabolic rates and the concentration of specific molecules. Brain PET imaging applies especially well to these metabolic and biochemical processes, which gives insights into glucose metabolic activity in the cells and neurotransmitter receptor density. In the underlying PhD project [14], these principles were applied to the neural activity measured by cellular glucose metabolism and dopamine receptor binding potential. 

#### 1.2.2. FDG-PET

The brain consumes approximately 20% of the body’s total glucose demand [16]. For the measurement of glucose metabolism, PET exploits the first step of glycolysis, the initial stage of producing energy in the form of ATP in the cells. Here, the enzyme hexokinase phosphorylates the glucose molecule, trapping it within cells and initiating its metabolism [17]. FDG is a glucose analogue labeled radiotracer with a radioactive fluorine isotope (18F) with a radioactive half-life of 109.8 min [18]. FDG acts as glucose but is not metabolized beyond hexokinase (Figure 2), which characterizes FDG as an irreversible tracer. 

When planning and designing an FDG-PET study, more elements have to be considered regarding the radiation properties and pharmacokinetics of the tracer. Once administered into the bloodstream, FDG is absorbed by tissues in proportion to the metabolic activity and reaches an equilibrium point, at which the distribution of FDG in the body reaches a relatively constant level in the exchange between the blood pool and tissue, approximately 30 to 60 min post-injection [2]. 

#### 1.2.3. Principles of MRI

MRI is a non-invasive imaging technique that uses strong magnetic fields and radio waves to generate detailed images of the brain’s internal structures and function [19]. A major contribution of MRI to neuroimaging is structural MRI, providing high-resolution images of anatomical features in the brain, for example gray and white matter. For structural anatomical images, MRI relies on the behavior of hydrogen atoms in the water molecules of the body, specifically the protons within them. In the scanner, the protons are initially in a state of equilibrium subjected to the main magnet field B0. The protons are then excited by radiofrequency pulses and the variations in relaxation times and signal intensities in different tissues allow generation of detailed images (Figure 3). The gradient coils vary the strength of the field along axes (X, Y, and Z), enabling precise spatial localization of signals.

### 1.3. Genotype Detection

The *COMT* gene, described as the Warrior/Worrier gene [20], codes for the catechol-O-methyltransferase enzyme, which breaks down dopamine in the prefrontal cortex. The wild-type G allele codes for valine (Val), whereas the A allele polymorphism changes the amino acid to methionine (Met). The altered composition causes the activity of the enzyme to be reduced to 25% of the wild type. As a result, A allele carriers have more synaptic dopamine in their prefrontal cortex [21], which is considered advantageous for prefrontal function and related executive and cognitive function (Figure 4) [22,23]. 

It is believed that the Val allele (Warrior) gives an emotional advantage relative to Met (Worrier) allele carriers [24], making Worriers less resilient to stress [25]. The *COMT* gene is known to be related to the onset of mood disorders after stressful life events [26].

## 2. Materials and Methods

This was a single-center, observational, comparative study. 

### 2.1. Study Population

All stress patients included in the study were recruited at the Department of Occupational and Environmental Medicine (DOE) at Odense University Hospital (OUH), Odense, Denmark. This department is one of the seven active departments in Denmark. Patients are primarily referred by General Practitioners, with a small percentage (less than 10%) being referred by other professionals such as trade unions, other hospital sections, or safety organizations at workplaces [27]. DOEs are part of the Danish welfare and health system and are accessible to all individuals free of charge. At DOE OUH, DK patients are referred to an occupational health psychologist and undergo a comprehensive investigation. This investigation focuses on evaluating their mental health, as well as assessing their lifetime and current stressors both within and outside of work, along with relevant etiological factors associated with psychosocial stress. The assessment concludes with a diagnostic evaluation and an appraisal of whether the mental illness can be considered an occupational disease (i.e., most likely the result of work-related exposure), a work-related disease (i.e., work exposure contributes to the disease but the disease is multifactorial), or a disease unrelated to exposures at work. Determining whether a stress-related disorder is predominantly caused by one factor or another is challenging. Here, all stress patients undergo assessment by trained psychologists, which includes screening with psychometric screening tools. Diagnosis is determined through a clinical interview that assesses the prodromal stages of the illness, with a focus on the type and extent of symptom complaints and their development over time. Current symptoms and level of functionality are also assessed, with a focus on general somatic, emotional, behavioral, and cognitive symptoms. Additionally, patients are asked about their current somatic diseases and their history of mental illness, both in themselves and their family members. Screening for depression, anxiety, and, when appropriate, post-traumatic stress disorder is also conducted. Here, the screening tools of the Major Depression Inventory (Depression) and Beck Anxiety Index (anxiety) are included in the screening process. Screening of anxiety and depression is mandatory in the assessment of stress, due to the high prevalence of depression and anxiety as common mental health disorders and their significant symptom overlap with diseases caused by severe and chronic stress.

The assessment of relevant etiological factors includes evaluating the patient’s history of mental illness, somatic diseases, adverse stressors during developmental phases (e.g., childhood abuse, neglect, violence, parental drug or alcohol abuse, loss of or serious illness in a parent or sibling), as well as indications of learning disabilities or severe problems in school (e.g., severe bullying). Previous or current drug or alcohol abuse, lifetime traumatic incidents in both childhood and adulthood, and chronic or serious diseases are also included in this assessment. Assessment of current stressors outside of work, for example serious illness of loved ones, financial problems, interpersonal problems, and divorce, is also conducted. 

Work exposure was assessed by screening for recognized stressors in the workplace, including screening for stressors in the psychosocial work environment (Appendix A), including quantitative demands (workload, time pressure, and high quantitative demands), emotional demands, role stressors (e.g., lack of role clarity or conflict, lack of predictability), low decision authority, low social and instrumental support, low meaning, low levels of justice (informative, procedural, or interpersonal), interpersonal conflicts, bullying, threats and violence, sexual harassment, and exposure to critical incidents [8,28]. The experience of stressors was assessed in relation to the most dominant stress models, *demand–control–support* [29], *effort and reward imbalance at work* [30], *Stress as offense to self* [31], and classical stress theory, focusing on appraisal/coping [32]. The overall appraisal of whether the work exposure can be considered the primary cause of stress development is based on several factors. These include the correspondence between the time of exposure and the onset of stress symptoms, the reasonable exclusion of primary competing stressors and serious etiological factors outside of work, whether the work exposure corresponds to known risk factors for stress-related disorders, and whether the process of stress development can be explained within accepted theories of work-related stress, such as the demand–control–support model, the effort–reward imbalance model, or the stress as an offense to self model. Only when these conditions are met is the illness considered a probable occupational disease. In DOEs, patients with stress-related disorders receive a diagnosis of Z56.x in cases where the mental health problems are not severe enough to be classified as psychiatric disorders. Patients with severe stress disorders receive a diagnosis of either F43.2 adjustment disorder or F43.9 reactions to severe stress [33]. There is some variation among the seven clinics in terms of how they distinguish between these diagnostic groups. At DOE OUH, the F43.2 adjustment disorder diagnosis is used for patients with occupational stress disorders resulting from general work stressors, such as quantitative and emotional demands, role stressors, and injustice. The F43.9 diagnosis is used for occupational stress disorders where the stressor is more severe, such as violence, bullying, sexual harassment, and critical incidents, and when the A-criteria for post-traumatic stress disorder are not met.

To account for gender differences in stress response and neurobiology, only women were included, who were 18–64 years of age, diagnosed with a F43.2 diagnosis, and identified as suffering from an occupational mental health disorder. During the trial period, there was an average waiting list interval of approximately 14 weeks at the DOE. The inclusion and exclusion criteria were the following:


**Inclusion criteria:**
Informed consent before study-related activity;Females aged 18–64;Labor market suitability;Patients: F43.2 according to ICD-10;Controls: Healthy individuals that met the inclusion and exclusion criteria.



**Exclusion criteria:**
Adjustment disorder other than work-related;Disorders specifically associated with non-work-related stress;Anxiety-related disorders;Mood disorders;Major comorbid diseases, for instance: cancer, cardiovascular disease, diabetes;Medication known to affect the central nervous system;Prior exposure to violence and other serious harassment in the workplace;Psychosocial challenge in private life according to screening criteria stated in the protocol;Other serious illness;Metal in the body not compatible with an MRI scan;Claustrophobia;Pregnancy;Confounding drug consumption;Dependence, e.g., alcohol, narcotics, or other.


Healthy control (HC) women were recruited via a wide variety of platforms, including the OUH Blood Bank and online on social media, and screened for exclusion criteria in a similar manner to the patient group. The patients and controls were comparable according to age. Both patients and HCs were currently active in the labor market, but patients were on sick leave. The menstrual cycle, menopause, and similar factors have not been accounted for as confounding variables in this study.

The participants were recruited during the period from September 2019 to November 2020. The scans were conducted within a few weeks after the first contact. 

### 2.2. The Schedule for Clinical Assessment in Neuropsychiatry (SCAN) Interview

The mental state of the participants was assessed using a SCAN interview immediately before the brain scans [34]. In the case of somatic comorbidities, eating disorders, psychosis, and mania the patients were excluded (Appendix A). 

### 2.3. PET/MR Image Acquisition and Pre-Processing

The study was conducted on a combined 3.0T SIGNA^TM^ PET/MRI (GE Healthcare, Chicago, IL, USA). Glucose metabolism was measured using FDG, while structural MRI images were acquired simultaneously during the PET scans. 

Patients and control subjects fasted for 4–6 h before injection of FDG. They then rested on a bed wearing a blindfold and earplugs in a quiet room for 10 min before tracer injection. The FDG tracer was administrated intravenously by an Intego PET infusion system as a single bolus injection, using a mean dose of 200 MBq of FDG for both patients and HCs. Afterward, the participants rested for another 40 min before walking to the scanner room, where they were placed supine in the scanner with their head immobilized in a dedicated headrest. All participants were scanned on the same scanner with a designated head/brain coil. List-mode PET data were acquired by a standard acquisition protocol for 15 min along with a simultaneously acquired T1-weighted (T1) structural MR for all participants, reconstructed into a single scan for each modality. The field-of-view (FOV) was 22 cm with a matrix size of 128 × 128. MR-based attenuation correction was performed with the clinical atlas-based method as implemented on the SIGNA PET/MR system, where individual T1 MR images were rigidly and non-rigidly registered to a CT-based head atlas. The PET data with native voxel dimension (2.78 × 2.78 × 2.78 mm^3^) were reconstructed using a fully 3D TOF iterative ordered subset expectation maximization algorithm with point spread function (28 subsets, 3 iterations), and corrected for attenuation, scatter, dead time, and decay. 

The FDG-PET data were analyzed using standardized statistical parametric mapping and processed in the SPM Version 12, released 1st October 2014 and last updated January 2020 (PET and VBM module) software analysis program with default parameters maintained, unless otherwise specified. At the single-subject level, the structural T1 images were co-registered to the FDG-PET images, followed by a segmentation of grey matter (GM), white matter (WM) and Cerebrospinal Fluid (CSF) based on the SPM12 tissue probability map and moved into the Montreal Neurological Institute (MNI) standard space. A forward deformation field was saved to utilize normalization of the data into the standard MNI space. FDG-PET images were then warped into the MNI space and smoothed with a 5 mm kernel. At the group level, analysis images were pooled for the HCs and the patients, respectively, and the SPM12 brain template was applied to the data. Group-level FDG images were tested for voxel-wise differences in FDG uptake between HCs and stress patients throughout the brain. Here, two-sided *t*-tests were applied to identify voxels for which there was a statistical group difference in FDG uptake, where stress patients showed either an increase or decrease in FDG uptake (i.e., HC > Patients or HC < Patients) at a voxel-level uncorrected *p*-value of 0.001 (see also Section 2.5).

### 2.4. Genotyping

Blood samples were collected from both stress patients and HCs for analyzing genotype Single-Nucleotide Polymorphism (SNP) variation in the *COMT* gene by quantitative polymerase chain reaction (qPCR).

DNA for genotyping was isolated and purified from whole-blood samples from the participants using the Qiagen QIAamp DNA Mini Kit (Qiagen, Valencia, CA, USA). The DNA concentration was determined with a Qubit^®^ dsDNA HS Assay (Thermo Fisher Scientific, Waltham, MA, USA). *COMT* Val158Met (rs4680) variations were determined via qPCR with a TaqMan SNP genotyping assay (C__25746809_50, Thermo Fisher Scientific). Amplification and detection were performed on a QuantStudio 3 (Thermo Fisher Scientific). Samples were run in triplicate with DNA from the breast cancer cell lines MCF7 and MDA-MB231 as controls.

### 2.5. Statistics

We hypothesized that the SCAN interview data of patients and HCs stem from the same distribution at the item question level (null hypothesis). Group differences were analyzed nonparametrically with the Wilcoxon rank sum test. The resulting *p*-values were sorted ascendingly and evaluated by means of the Bonferroni–Holm procedure [35]. The level of significance was 5%. All analyses were performed with STATA/IC 17 (Stata Corp, College Station, TX, USA).

In SPM12, two-sample design matrices were applied to test for voxel-wise differences in FDG uptake between HCs and stress patients throughout the brain. Here, one-tailed *t*-tests were performed to evaluate voxels for which there was a directional-specific group difference, i.e., HC > stress or HC < stress. To generate voxel-level statistical inference analysis in SPM results, no family-wise error (FWE) was set. SPM’s default uncorrected *p*-value (*p* = 0.001) was applied for the probability of a false positive at each voxel with no cluster extent threshold. The genotype detection analysis was based on nominal data of qualitative properties. The distribution is illustrated by means of descriptive statistics due to the small study population, that is, bar plots and frequencies with respective percentages. 

### 2.6. Ethical Considerations

Given the properties of the brain images acquired in the project, the chance of encountering undiagnosed pathologies was accommodated with a clinical inspection of the acquired data. All participants’ FDG scans were reviewed by a nuclear medicine specialist with expertise in brain imaging. In the case of abnormalities, the individual would be called in for further clinical examinations. Similarly, structural MRI scans were assessed by technical personnel; in cases of uncertainty, a radiologist with neurological expertise was involved, following the same procedure as with FDG. Conclusively, none required further investigation or treatment.

## 3. Results

In total, 11 patients and 11 HCs, all female, were included in the study, with a mean age of 48.1 years (range: 30–62 years) and 42.8 (range: 28–55 years), respectively. The scans were assessed by a medical doctor specialized in nuclear medicine and no individual pathological variances of importance were identified.

### 3.1. SCAN Interview

Eleven stress patients and 10 HCs were included in the SCAN interview analysis, as the dataset was not complete for one participant. The mean value of ratings of the stress patients and the HCs, respectively, were calculated in each question and applied in the statistical analysis. In 18 out of the 159 questions in the interview, there was a statistically significant difference between the stress patients’ and the HC’s ratings of the perception of the state of their mental health situation and somatic symptoms of stress (Appendix A). 

### 3.2. FDG-PET Brain Imaging Scans

Due to data loss on the scanner, only 8 HCs and 10 stress patients were included in the brain imaging analysis. The one-tailed *t*-tests performed to evaluate voxels for which there was a directional-specific group difference (i.e., HC > stress or HC < stress) only showed effects in the HC > stress group. The one-tailed *t*-tests evaluating voxels for which there was a directional-specific group difference driven by a hypometabolic effect in stress patients compared to HCs identified three clusters with MNI coordinates [−30 24 24], left frontal lobe sub-gyral adjacent to middle frontal gyrus, [28 8 44] right frontal lobe middle frontal gyrus, and [12 −16 68] right frontal lobe precentral gyrus. Noticeably, the identified cluster appeared primarily to be located in the WM regions (Figure 5).

Mean FDG uptake values were extracted from suprathreshold clusters for visualization from each participant (Appendix A). Value distributions are illustrated in the box plots (Figure 6).

### 3.3. Genotyping

Eleven patients and 11 HCs were included in the genotyping of the SNP variation in the *COMT* gene. The result of the qPCR analysis is illustrated in (Figure 7).

The homozygous Warrior allele Val/Val amounted to two (18%) in the patient group and one (9%) among the HC group; the heterozygous allele Val/Met amounted to fiive (46%) in the patient group and nine (82%) among the HC group, and the homozygous Worrier allele Met/Met amounted to four (36%) in the patient group and one (9%) in the HC group.

## 4. Discussion

### 4.1. Principal Findings

The results of the adapted version of the SCAN interview showed a clinical direction of stress patients experiencing physiological dysfunction as well as mental challenges in higher brain functions related to executive functions and emotion and mood regulation. Our FDG-PET imaging group analysis identified three clusters of voxel-wise differences in cerebral glucose metabolism between stress patients and HCs, where the stress patients had hypometabolic effects in areas primarily located in the white matter of the prefrontal cortex. The extracted suprathreshold cluster mean FDG uptake values for each participant expresses a consistent difference in the distribution between stress patients and HCs. 

The genotype detection did not give a clear indication of the carriers of the homozygous Worrier variant being less resilient to psychosocial stressors in the work environment, whereas the HCs seemed to be pooled in the heterozygous variant group, with a notion about the European population *COMT* gene variation frequencies differing compared to other parts of the world.

### 4.2. Strengths and Limitations

The small sample size is a limitation. However, a decreased signal from WM was observed, suggesting a possible hypometabolic effect in stress patients. If this is indeed the case, it should be noted that a relative reduction in signal related to FDG uptake in WM voxels compared to a signal originating from GM voxels would require a larger actual loss of neural activity, as the density of neurons is lower in WM and hence the signal more exposed to noise. Furthermore, we only included female subjects, which means that our results cannot be generalized to males. This, however, makes our sample more homogeneous. Another limitation is the lack of international standardization of an independent stress diagnosis, especially work-related, and the evident overlap with depression and anxiety. As an alternative, we chose to implement the SCAN interview, which made it possible to characterize work-related stress in a reproducible way and thus served as an effective tool for the investigation of the general state of health of stress patients. The severity of the symptoms varies within as well as between subjects in patients with stress. 

As genotype frequencies appear to vary globally, this could potentially affect the distribution of stress resilience in populations based in Europe compared to other continents. 

### 4.3. Relation to Other Studies

To the best of our knowledge, no study of the neurobiological effects of documented work-related stress by means of FDG-PET has been conducted prior to this study. In comparison, depression and anxiety, which both share common symptoms with stress, have been studied to a greater extent. From a voxel-based meta-analysis of FDG-PET, it is evident that depression produces significant changes in FDG uptake in the brain, causing decreased metabolism in the insula, limbic systems, and basal ganglia, but increased FDG uptake in the thalamus and cerebellum in patients with major depressive disorder compared to healthy controls [36]. A later study assessing resting state cerebral function with FDG-PET in depression likewise found hypometabolism and hypofunction in regions of the frontal and temporal lobe, the bilateral superior, and the middle and inferior frontal gyrus, but hypermetabolism in the hippocampus and left thalamus. Changes in frontal lobe function are believed to play an important role in the neurobiology of major depression [37]. Similarly, in relation to anxiety, FDG-PET scans comparing HCs with patients with generalized anxiety disorder found evidence of reduced metabolism in brain areas related to emotional control and affective regulation [38]. Our exploratory results of changes in brain metabolism related to work-related stress seem to be in alignment with findings in depression and anxiety, as the trend in our findings shows similar signs of bilateral hypoactivity in the middle frontal gyrus [39]. 

The literature regarding the study of WM hypometabolism by the means of FDG is somewhat limited [40]. MRI studies show that significant thinning of the mesial frontal cortex [41] and hypometabolism in WM could be connected to disturbances in the structural connectivity of nerve fiber tracts in WM [42]. It is suggested that the pathology of WM hypometabolism may be related to neuroglia and axonal function [43]. WM tractography’s correlation with glucose metabolism can be studied by the means of combined FDG and MR diffusion tensor imaging (DTI) [44].

Besides the nerve cell, supporting glia cells play essential roles in supporting and maintaining the proper functioning of neurons and the overall health of the nervous system. Astrocytes provide structural support to neurons by forming a network of processes that surround and maintain the positioning of neurons in the brain, and they are involved in forming and maintaining the blood–brain barrier. Astrocytes also regulate the levels of neurotransmitters in the synaptic cleft and help maintain the balance of ions, such as potassium and sodium, in the extracellular fluid surrounding neurons, ensuring proper nerve cell function that supports the survival and growth of neurons, aiding in their health and promoting the formation of new synapses. Notably, astrocytes outnumber neurons in the human brain and are active in white matter [45,46]. Oligodendrocytes are responsible for myelinating axons. Myelin is the fatty substance that wraps around axons, speeding up the transmission of electrical signals and insulating the nerve fibers [47]. Microglia are the immune cells and brain-resident immune system and are responsible for detecting and responding to pathogens, injury, or inflammation in the brain. They also play a role in removing damaged cells and debris [48]. As a whole, glia cells help regulate the exchange of nutrients, ions, and other substances between blood vessels and neurons. They facilitate the transportation of essential molecules to neurons and assist in the removal of waste products. They release factors that influence neuron health, and the formation of new synapses, particularly astrocytes, is involved in regulating synaptic activity by modulating neurotransmitter release and clearance, influencing synaptic plasticity and overall brain function. 

The perspective that brain disorders are in fact related to more than neural deficiency, and the neurobiological effects of work-related stress, caused by a number of biochemical and cellular mechanisms, broadens the horizon for how the brain processes glucose; hence, the association between stress and abnormal glucose metabolism, insulin resistance, and metabolic syndrome in a wider perspective, closely related to biochemical energy production, and also metabolic syndrome, has been addressed in more stress study populations. Among others, a study of more than 6000 participants, assessed with the COPSOQ to evaluate occupational stress, suggested it to be an independent risk factor for developing insulin resistance and type 2 diabetes [49]. A close association between job-related stress, assessed with the COPSOQ, and the presence of metabolic syndrome, which is also an insulin resistance-related syndrome, could be another indicator of metabolic energy deficiencies and gender differences [50,51] as well as perceived psychosocial stress predicting an abnormal glucose metabolism [52]. This suggests that stress is correlated with impaired glucose metabolism. When the body is subjected to high levels of work-related stress, there is a continuous secretion of cortisol. This ongoing release of cortisol directly interferes with the cellular absorption and utilization of glucose, resulting in reduced insulin sensitivity and an increased presence of insulin resistance within the body [53].

So, in addressing the results of this study and the complementary results from the literature, a stress-related decrease in glucose metabolism in white matter, gray matter reduction, downregulated functional connections, etc., could very well be multifactorial and possibly caused by glia cell dysfunction, since white matter contains a significantly higher glia-to-neuron ratio than gray matter [54].

As a whole, ‘Brain Energetics” seems an emerging and promising field of research of brain disorder pathologies, but is potentially also an area of therapeutic and pharmacological interest [55]. Popular medications, such as GLP-1 agonists that work by stimulating the release of insulin and lower blood sugar levels, are currently also being investigated in relation to stress [56].

Regarding our results on genotyping, they do not appear to support the theory of the Val/Val Warrior variant being more stress resilient than the Met/Met Worrier variant. In fact, the genotypes of the stress patients included in this study are fairly equally represented by the homozygous genotype as well as the heterozygous one although actually with a fewer of the Met/Met genotype. One could argue, however, that work-related stress in the modern labor market can be related to emotional stressors as well as to pressures associated with the ability to perform executive functions. This is consistent with the Val/Val variant showing a modest reduction in executive cognition performance and the Met/Met variant showing enhanced vulnerability to emotional stress [57]. For the HCs, the picture is even more ambiguous as the vast majority of the participants in our group are in the heterozygous Val/Met variant group. Therefore, the pathology and symptoms of the stress patients included in this study are not directly comparable to earlier findings, which indicates that Met allele genotypes are less able to process emotional stimuli and more likely to have an elevated risk for emotion-related psychopathology [22]. The latter includes a reduced ability to shift attentional focus away from aversive stimuli and hence an impaired ability to disengage attentional resources from perceived sources of threat [58]. 

Studies of major depression and anxiety show a similar inconclusiveness concerning the distribution of the *COMT* gene polymorphism. In a review of the role of *COMT* gene variants in depression, most studies focusing on the difference between depressed patients and HCs did not detect significant results [59]. Some variations were described in association with the Met/Met genotype and depression and between the Val/Val type and early onset depression [26,60]. The non-significant findings seem to apply to *COMT* gene variation in anxiety as well as depression [61]. In this vein, a non-significant correlation in the frequency of the *COMT* polymorphism was found between patients with generalized anxiety disorder and HC groups, as well as between patients with different genders [62]. However, stress seems to be a key factor in the etiology of major depressive disorder [63], and the *COMT* genotype could possibly enhance the predisposition to depression by altering the reactivity to stressors [64]. Furthermore, population frequencies of the Met/Met and Val/Val variants seem to vary globally with an almost equal distribution in Europeans, while the Val/Val variant is more common in populations elsewhere in the world [65]. Hence, the phenotype of disorders such as stress, depression, and anxiety seems complex with a multifactorial etiology. A realistic assumption from this perspective is, of course, that potential interactions or influences between multiple genes are related to stress resilience, as determined in other areas of psychiatry, which should be considered in future research [66,67].

## 5. Conclusions

With work-related stress being a major and growing challenge in modern societies, our findings should raise cause for reflection and further investigations. The stress patients participating in this project have all been exposed to psychosocial stressors at work, which was conclusively the main factor in the cause of their stress condition. Further studies with larger samples are needed to research these perspectives. If access to a PET/MR scanner is available, a combined study applying DTI would be preferred to assess the pathology of WM abnormalities. The biochemical and physiological kinetics of stress response is a vast and multifunctional system, ranging from the molecular level of many cellular components to the conscious interpretations of one’s own situation by the individual with the brain as the main mediator [68]. This entails a network of interconnected neural components that work together to synchronize an individual’s behavior with their neuroendocrine, immune, and autonomic functions, all in support of effectively managing environmental and psychosocial stressors [69,70]. We found no indication of a correlation between *COMT* genotype variation and stress resilience, but we suggest further investigations assessing executive functions through a subdivision of neuropsychological tests addressing, for example, set-shifting, working memory, and inhibition ability tasks [71]. The grasp on and parameters of the understanding of stress and attempts to measure the neurobiological effects of the condition will be ongoing and continuously investigated for years to come.

## Figures and Tables

**Figure 1 diagnostics-14-01730-f001:**
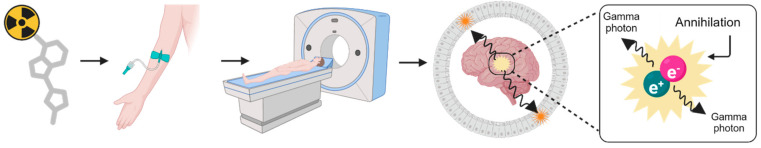
Principles of the PET modality.

**Figure 2 diagnostics-14-01730-f002:**
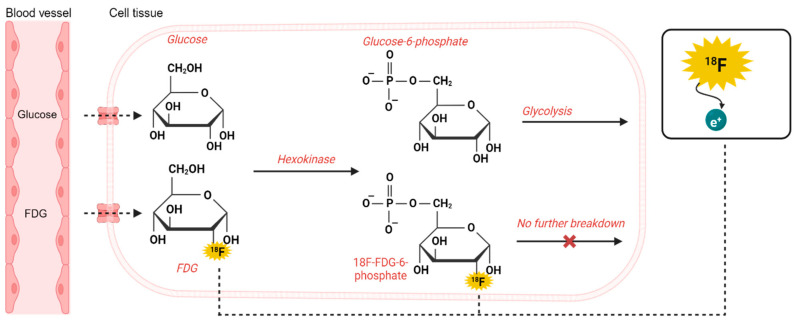
Principles of glucose and FDG metabolism.

**Figure 3 diagnostics-14-01730-f003:**
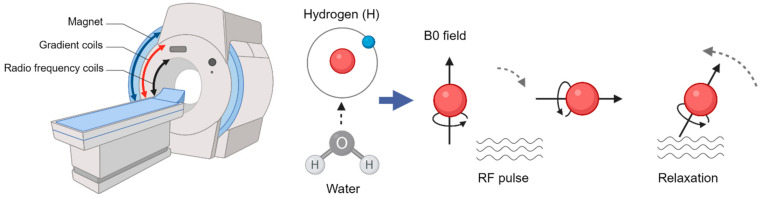
Principles of the MRI modality.

**Figure 4 diagnostics-14-01730-f004:**
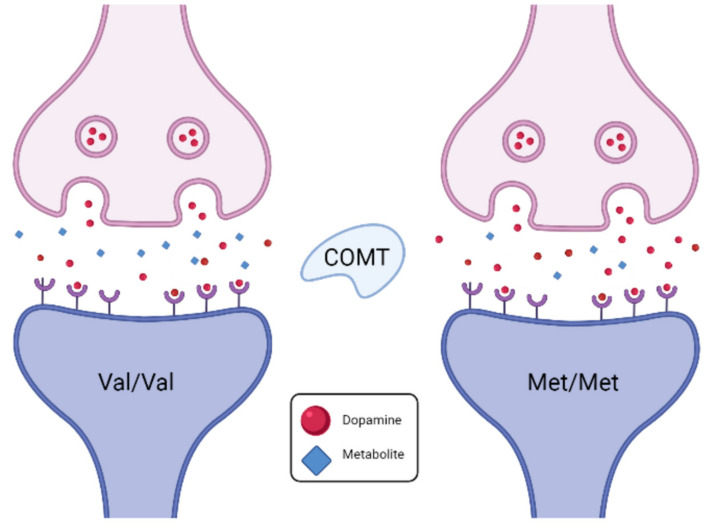
Principles of *COMT* genotype variations in synaptic dopamine availability.

**Figure 5 diagnostics-14-01730-f005:**
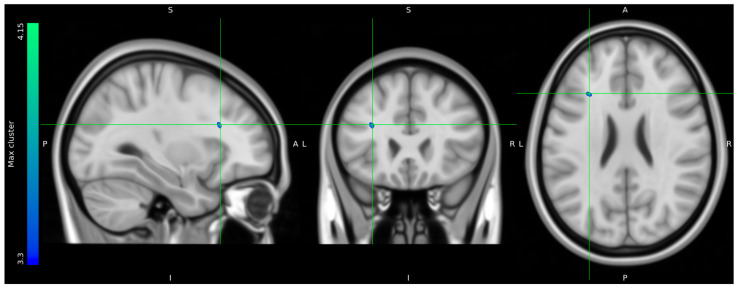
FDG uptake illustrating a difference in glucose consumption between stress patients and healthy controls. Here, projected on MNI standard space template MNI152 0.5 with intensity threshold defined in color bar on the left. A: anterior, P: posterior, S: superior, I: inferior, L: left, R: right.

**Figure 6 diagnostics-14-01730-f006:**
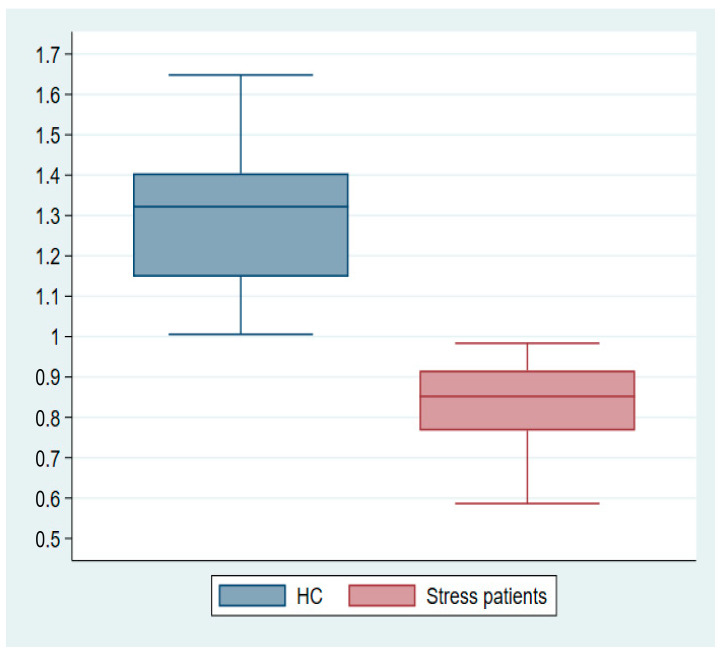
Mean cluster estimates subtracted from single-subject glucose consumption in the maximum cluster of the hypometabolic effect *t*-test.

**Figure 7 diagnostics-14-01730-f007:**
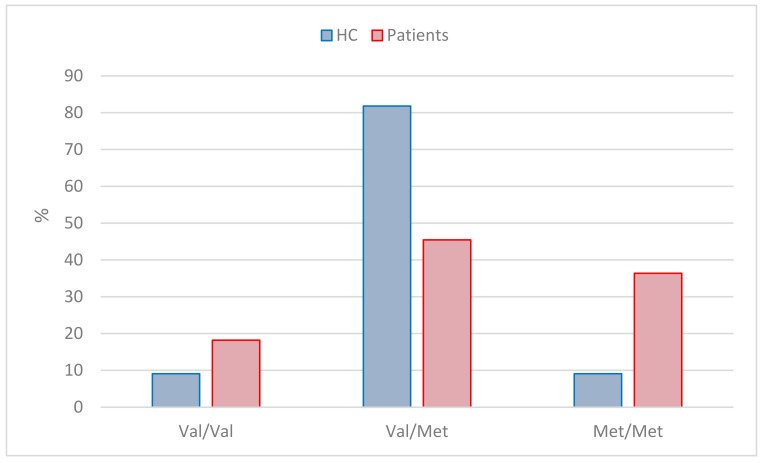
Illustration of genotype distribution between stress patients and HCs.

## Data Availability

The data of this study are unavailable due to legal restrictions.

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
