# Peer review of "Brain Glucose Metabolism and COMT Val 158 Met Polymorphism in Female Patients with Work-Related Stress"

_diagnostics, 2024, doi:10.3390/diagnostics14161730_

Round 1
Reviewer 1 Report
Comments and Suggestions for Authors
This study titled "Brain Glucose Metabolism and COMT Val158Met Polymorphism in Female Patients with Work-related Stress" by Madsen et al. presents an investigation into the neurobiological effects of work-related stress, particularly focusing on cerebral glucose metabolism and the COMT Val158Met polymorphism. This study bridges a gap in research by linking psychosocial work stressors to alterations in brain energy metabolism. [18F]fluorodeoxyglucose PET imaging provides a robust method to measure brain glucose consumption directly related to neural activity. Including PET and MRI imaging modalities allows for a detailed brain structure and function analysis. The SCAN interview thoroughly assesses the participants' mental health, ensuring a reliable diagnosis of work-related stress. The exploration of the COMT Val158Met polymorphism adds a genetic perspective to the study, which is crucial for understanding individual differences in stress resilience. This study identifies significant clusters of decreased glucose consumption in the white matter of the frontal lobe in stress patients. This finding supports the hypothesis that work-related stress impacts brain metabolism, particularly in regions associated with higher cognitive functions.
However, like other studies, this study has limitations and requires revisions to improve its comprehensiveness and scientific rigor.
1. This study's sample size is relatively small, with only 11 stress patients and 11 healthy controls. This limits the generalizability of the findings and necessitates further research with larger populations to validate the results. Additionally, the study includes only female participants, which, while making the sample more homogeneous, limits the applicability of the findings to males. It would benefit the authors to explain the rationale for selecting only female participants, perhaps considering factors such as gender differences in cognitive function or stress response. Additionally, it suggests in the conclusion that future studies include male participants to explore potential gender differences in stress-related brain metabolism, which would enhance the comprehensiveness of the research. The following references might provide some insights:
(1) Vigna, Luisella, et al. "Determinants of metabolic syndrome in obese workers: gender differences in perceived job-related stress and in psychological characteristics identified using artificial neural networks." Eating and Weight Disorders-Studies on Anorexia, Bulimia and Obesity 24 (2019): 73-81; (2). Chen et al. "The impact of sex on the neurocognitive functions of patients with Parkinson’s disease." Brain Sciences 11.10 (2021): 1331.
2. The scanner's data loss prevented the analysis of only a subset of the participants' brain imaging data, further reducing the study's statistical power.
3. The genotyping results do not clearly indicate the relationship between COMT genotype and stress resilience. The distribution of genotypes in the study sample might not reflect the broader population, limiting the conclusions about the genetic underpinnings of stress resilience. Additionally, the authors might consider exploring different linear relationships and discussing these further to enhance the richness of the content. For example, see Fang et al., (2019). More than an “inverted-U”? An exploratory study of the association between the catechol-o-methyltransferase gene polymorphism and executive functions in Parkinson’s disease. PLoS One, 14(3), e0214146. )
4. Given that this study focuses on a single gene, it's important to note that potential interactions or influences between multiple genes should not be overlooked. (e.g., Taheri, Narges, et al. "Association of DRD2, DRD4 and COMT genes variants and their gene-gene interactions with antipsychotic treatment response in patients with schizophrenia." BMC psychiatry 23.1 (2023): 781.; Yu et al., (2021). Interactions of COMT and ALDH2 genetic polymorphisms on symptoms of Parkinson’s disease. Brain Sciences, 11(3), 361.)
This consideration could significantly enhance the depth and breadth of future research on this topic, thereby increasing the value of the current study's findings.
Based on the above comments, this study addresses an intriguing research topic. It is recommended that the authors revise the article to enhance its completeness.
Author Response
Please find our response letter attached as PDF, thanks.

Reviewer 2 Report
Comments and Suggestions for Authors
The manuscript entitled:„ Brain Glucose Metabolism and COMT Val 158 Met Polymorphism in Female Patients with Work-related Stress „ is a single-center, observational comparative study examining the link between work-related stress and the neurobiological element of brain processing. The hypothesis states that patients with occupational stress have altered cerebral glucose consumption compared to healthy controls.
Major comments:
1. The main drawback of the study is the small number of subjects in both groups to make any conclusion about work-related stress, which is too complex to make a clear conclusion about glucose metabolism in the brain.
2. Additionally, the number of subjects in each of the subgroups for the Val and Met polymorphisms is insufficient for any conclusion about the association of the gene with depression. Therefore, practically nothing can be concluded from Figure 7, and does not represent any important scientific fact for research.
3. There are several studies related to Val Met polymorphism, fMRI, and depression that include a significantly larger group of subjects (over 100 subjects). Therefore, the manuscript was written with unnecessary descriptions, and generalized facts, without referring to relevant previous scientific research related to MRI, Val Met polymorphism, and emotions.
4. The authors did not describe or explain the expression and role of COMT in the human brain, as well as its variants, linking them to previous published studies like: http://www.ncbi.nlm.nih.gov/pubmed/17008817)Unnecessary description of the working principle of the MR device and the working principle of PET (Principles of PET) - these are well-known facts that should not be explained.
5. Unnecessary description of the working principle of the MR device and the working principle of PET (Principles of PET) - these are well-known facts that should not be explained.
6. The text related to the Principles of glucose and FDG metabolism is also unnecessary.
7. Why were only women chosen? A wide range of ages of the respondents, most of them in the reproductive age - the influence of the menstrual cycle?
8. In the materials and methods, it is not separated what are the inclusion and exclusion criteria of the respondents.
9. It should be clearly emphasized what are the clear criteria for the selection of respondents - based on which exactly the respondent was chosen as work-related stress.
10. It is also not clear whether the control group was also all women or not.
11. It was stated that the patients were recruited from 2019 to 2020, which is 2-3 years ago, so the results are not exactly the latest.
Minor comments
1. COMT gene - meaning of abbreviations? (in abstract - line 28)
2. Why is Occupational Stress written in the keywords, and work-related stress in the title.
3. It was stated that the patients were recruited from 2019 to 2020, which is 2-3 years ago, so the results are not exactly the latest.
4. The meaning of the abbreviation MNI coordinates - line 351
Author Response

(The authors gave the same response as above.)

Round 2
Reviewer 2 Report
Comments and Suggestions for Authors
The manuscript entitled " Brain Glucose Metabolism and COMT Val 158 Met Polymorphism in Female Patients with Work-related Stress " was thoroughly revised by the authors based on the feedback provided by the reviewers. The authors changed the manuscript's conception and conclusion according to my suggestions. They have mentioned scientifically proven facts in the new text and explained the advantages of PET as an important method for measuring glucose consumption and brain activity. They used the glucose analog radiotracer FDG, with MRI providing anatomical and structural overlay. They explained the reasons for the small sample size of their study (such as COVID-19 restrictions).
Due to the improved writing, the manuscript is now suitable for publication and should be considered for acceptance.